# Genetic Mechanism Study of Auditory Phoenix Spheres and Transcription Factors Prediction for Direct Reprogramming by Bioinformatics

**DOI:** 10.3390/ijms231810287

**Published:** 2022-09-07

**Authors:** Jishizhan Chen, Ziyu Liu, Jinke Chang

**Affiliations:** 1UCL Centre for Biomaterials in Surgical Reconstruction and Regeneration, Division of Surgery & Interventional Science, University College London, London NW3 2PF, UK; 2Beijing Advanced Innovation Centre for Biomedical Engineering, School of Engineering Medicine, Beihang University, Beijing 100083, China

**Keywords:** hearing loss, auditory neurons, phoenix cells, bioinformatics, gene expression, RNA sequencing, direct reprogramming

## Abstract

Background: Hearing loss is the most common irreversible sensory disorder. By delivering regenerative cells into the cochlea, cell-based therapy provides a novel strategy for hearing restoration. Recently, newly-identified phoenix cells have drawn attention due to their nearly unlimited self-renewal and neural differentiation capabilities. They are a promising cell source for cell therapy and a potential substitute for induced pluripotent stem cells (iPSCs) in many in vitro applications. However, the underlying genomic mechanism of their self-renewal capabilities is largely unknown. The aim of this study was to identify hub genes and potential molecular mechanisms between differentiated and undifferentiated phoenix cells and predict transcription factors (TFs) for direct reprogramming. Material and Methods: The datasets were downloaded from the ArrayExpress database. Samples of differentiated and undifferentiated phoenix cells with three biological replicates were utilised for bioinformatic analysis. Differentially expressed genes (DEGs) were screened and the Gene Ontology (GO) terms and Kyoto Encyclopedia of Genes and Genomes (KEGG) pathway enrichment were investigated. The gene set enrichment analysis (GSEA) was conducted to verify the enrichment of four self-defined gene set collections, followed by protein-protein interaction (PPI) network construction and subcluster analysis. The prediction of TFs for direct reprogramming was performed based on the TRANSFAC database. Results: Ten hub genes were identified to be the key candidates for self-renewal. Ten TFs were predicted as the direct reprogramming factors. This study provides a theoretical foundation for understanding phoenix cells and clues for direct reprogramming, which would stimulate further experiments and clinical applications in hearing research and treatment.

## 1. Introduction

Over 1.5 billion of the world’s population are living with hearing loss, due to unhealthy hearing habits, administration of ototoxic drugs and ageing, etc. [1]. Sensorineural hearing loss (SNHL) is the most common form of hearing loss which is typically caused by the degeneration of hair cells and auditory neurons (ANs) [2,3]. However, there are very limited clinical treatments for such profound hearing loss as SNHL [4]. Cochlear implant prosthesis is the most effective clinical routine for SNHL, which could partially relieve the hearing loss symptoms by sending electrical stimulation to the healthy population of ANs [5,6,7]. The invasive implantation cannot restore the cellular degeneration in the cochlea, but may cause further iatrogenic damage to remaining hair cells and spiral ganglion neurons (SGNs), leading to inflammatory responses, nerve fibre regression and AN degeneration [8,9,10]. Although many efforts have been made to improve the technique of cochlear implants, there are still many issues including the risk of device failure, gradually decreased auditory performance and diminished active electrode numbers [11,12].

The cell-based replacement and regeneration of ANs stand for a novel therapeutic strategy for restoring hearing functions, which has been extensively investigated with promising results. It also provides a biological approach to exploring hearing-related disease models, drug testing and pathogenic mechanisms. Several studies have explored various embryonic stem cells (ESCs) that can be successfully differentiated into sensory hair cells [13] and auditory neurons [14]. Koehler et al. [15] published an in vitro study that used murine ESCs to initiate stepwise differentiation of both inner ear hair cells and neurons. Human ESCs have also been investigated to differentiate into sensory neurons, with a gene expression profile similarity of up to 95% compared with native SGNs [16]. The ethical considerations and immuno-incompatibility of nano-autologous transplantations associated with hESCs has been a long-standing issue [17], therefore, the use of autologous stem cells is intensively explored. Autologous stem cells such as induced pluripotent stem cells (iPSCs) and bone marrow-derived mesenchymal stem cells [18,19] have also been used for auditory neuron models for the potential application of repairing and replacement of damaged ANs. iPSCs have provided a very promising strategy which can be generated by direct reprogramming of somatic cells and differentiated into many cell types. Many researchers have reported induction protocols to differentiate human iPSCs into AN-like cells based on different cell sources such as fibroblasts [20,21,22] and urinary cells [23,24,25]. However, there are still many pitfalls that have to be surpassed before using iPSCs for hearing research and hearing loss treatment, including cell ageing [26] and direct reprogramming efficiency [27,28]. Additionally, in order to obtain a sufficient number of cells, repetitive induction processes of iPSCs are required which might introduce more variations. There is also significant variability existing between human iPSC cell lines, limiting the large-scale generation [29]. The expenses and time-consuming process of iPSC-derived ANs have highly limited their application in the research of cochlear pathophysiology and in vitro hearing models [17].

Recently, phoenix spheres have attracted much attention due to their robust self-renewal properties, and direct reprogramming capability. Researchers have demonstrated an in vitro phoenix auditory neuroprogenitor model that exhibited robust intrinsic self-renewal properties beyond 40 passages. The phoenix spheres can be differentiated into SGNs efficiently at any passage [30]. It has also been demonstrated the self-propagation potential of phoenix sphere population that can be frozen and thawed on demand [31]. These unique properties have made phoenix cells a suitable cell line for in vitro hearing disease modelling and basic gene research. The high cell yield and low variability of the phoenix sphere provide a significant possibility of large-scale generation of AN-like cells in the lab. The direct reprogramming technique using transcription factors (TFs) may convert other cell types into phoenix cells [32], which achieves the ultimate goal of reducing the use of laboratory animals as cell sources. In terms of clinical application potential, implanting progenitor cells into the cochlea to replace damaged hair cells and auditory neurons is a logical strategy [33]. Although iPSCs have been extensively explored in this strategy, they sometimes display unexpected or uncontrollable outcomes such as tumorigenicity [34,35]. Different from the iPSCs, the phoenix cells are auditory progenitors and only commit to neural differentiation, which ensures safety. However, the genetic mechanism of the self-renewal and direct reprogramming of phoenix spheres is still not clear. In this paper, we used bioinformatic tools to investigate the genetic mechanism of the self-renewing capability and the reprogramming factors for AN-like cells.

## 2. Results

A total of 6 samples and 27,179 genes were uploaded to the NetworkAnalyst 3.0 for analysis, from which 8299 genes with constant values were removed. The remaining 18,880 genes were then mapped with the build-in Mus musculus reference genome. A total of 18,416 (97.5%) mapped genes were utilised for downstream analysis.

### 2.1. Identification of DEGs

The principal component analysis (PCA) plot (Figure 1a) shows that the phoenix spheres and the mature auditory neuron can be well separated into two populations, suggesting they have distinguishable characteristics between each other. The quantile normalisation plot (Figure 1b) displays that normalised gene expression data has similar medians, which indicates the data quality of each sample is comparable and suitable for the following analyses. A total of 1055 DEGs were identified, including 303 (28.7%) up-regulated genes and 752 (71.3%) down-regulated genes in the phoenix sphere group, compared to the auditory neuron. The volcano plot (Figure 2) highlights all up- and down-regulated DEGs over the thresholds of *p* < 0.05 and |log2FC| > 2.0. The heatmap (Figure 3) demonstrates both top 25 up- and down-regulated DEGs in the phoenix sphere group, compared to the auditory neuron group, showing great differences in the gene expression patterns between these two types of cells. Above all, the results indicate that the DEGs are reliable and can be used for the following analysis.

### 2.2. GO Terms and KEGG Pathways Enrichment Analysis

GO terms and KEGG pathway enrichment analysis of DEGs provide a comprehensive view regarding the biological characteristics of phoenix spheres. Figure 4 shows that the biological process of phoenix sphere is highly enriched in the cell cycle- and DNA repair-related GO terms, including cell cycle phase, mitotic cell cycle, cell division, and DNA replication. The cellular components are enriched in the chromosome, chromosomal part, and ribonucleoprotein complex. The structural constituent of the ribosome, structure-specific DNA binding, and structural molecule activity is enriched in the molecular functions. In the KEGG pathway enrichment analysis, the ribosome, cell cycle, and DNA replication are the most enriched pathways. A list of the top five terms in BP, CC, MF, and KEGG is shown in Table 1.

### 2.3. Gene Set Enrichment Analysis

The self-define collections initially contained 50, five, six, and four gene sets in the collections of cell cycle, auditory stimulus response, mechanical stimulus response, and electrical stimulus response, respectively. After trimming down the gene sets of too small or too large number of genes, it remained 30, three, six, and two gene sets in the aforementioned collections, which were utilised for further analysis. The results show that at the cut-off criterion |NES| > 1, nominal *p* < 0.01, and FDR q-value < 0.25, there is no statistical significance between the phoenix spheres and the auditory neurons in the collections of auditory, mechanical, and electrical stimulus response. In comparison, 27 out of the 30 gene sets in the cell cycle collection are significantly up-regulated in the phoenix sphere group (Figure 5).

### 2.4. PPI Network Analysis and TFs Prediction

The PPI network constructed by all DEGs includes 823 nodes and 3522 edges (Figure 6). Based on the connection degree calculated by the cytoHubba (Table 2), the Fn1, Gapdh, Mmp9, Vegfa, Itgb3, Cdk1, Col1a1, Met, Fgf2, and Aurka have the highest top 10 degree and thus are hub genes in the network. The Fn1 displays the highest degree (degree = 96), followed by Gapdh (degree = 91) and Mmp9 (degree = 54). The deletion of these 10 genes will remarkably affect the structure of the network and apply a negative effect on the interaction between proteins. Hence, these 10 genes are hub genes and play an important role in regulating the biological functions of the phoenix spheres.

### 2.5. Subcluster Analysis and Transcription Factors (TFs) Prediction

There are 31 biofunctional clusters in total identified by the MCODE, which suggests that the phoenix spheres obtain their characteristics through manoeuvring highly complicated biological modules. The top five clusters include nine out of the 10 hub genes (Figure 7). Detailed information of the top five clusters is shown in Table 3. Since the hub genes are mainly distributed in the cluster 1 to 3, the three clusters contribute more impact on the phoenix spheres. Therefore, the enrichment analysis focused on the top three clusters. The results illustrate that the cluster one is enriched in cell cycle-related terms. The top three BP GO terms are cell division, mitotic cell cycle, and mitotic cell cycle process. Cluster 2 and 3 are associated with cellular organisation, including axoneme assembly, microtubule bundle formation, and axonemal dynein complex assembly in cluster 2, and extracellular matrix organisation, external encapsulating structure organisation, and extracellular structure organisation in cluster 3. Detailed enrichment analysis results are shown in Figure 8 and Table 4. To reveal the potential TFs that can be used to convert other cell types into the phoenix spheres, a total of 823 proteins from the main PPI network were utilised to generate TF prediction. A total of 66 TFs were discovered, and the top 10 enriched TFs are highlighted in Figure 9. Detailed information is listed in Table 5.

## 3. Discussion

Normally, neurons have a very obvious self-renewal barrier and rapidly reach senescence after a few passages. However, phoenix spheres exhibited robust status even after 40 passages. In this study, bioinformatic analyses revealed 1055 DEGs in the undifferentiated phoenix spheres compared to the differentiated mature auditory neurons. The DEGs wove a huge interaction network of BP GO terms. Compared to the original report, although we used a narrower threshold to identify DEGs, our findings validated the previous report. For example, the GO term of the cell cycle and DEGs of Fa2h, Itgb4, Lgi4, Ntrk2, Miat, and so forth were also enriched in our data. In the BP network of this study, the cell cycle, DNA repair, metabolic regulation, and cellular organisation are the four main term categories comprising the main body. It is possible that the phoenix spheres have extremely high activity in cell division and a strong capability of manipulating cell components and structure. Additionally, nine out of the top 10 BP GO terms concentrate within the two term categories of cell cycle and DNA repairing, which suggest cell cycle and DNA repairing may be dominant processes that contribute to the nearly unrestricted self-renewal capacities of phoenix spheres. The phoenix spheres are originated from the A/J mouse, a commonly used lung carcinogenesis animal model [36]. There is a reasonable point that the exceptional self-renewal capacities are derived from carcinogenesis and an unstable genome. Nevertheless, the authors of the RAW data have proved that the phoenix spheres kept a normal cell cycle without mutated genomes even at a high passage number. In line with this, the auditory neuron differentiated from phoenix spheres also displays a significantly stable phenotype over passages [30]. Noticeably, the nearly unlimited self-renewable capabilities were merely observed in vitro. A/J mouse, the source of phoenix spheres, would still undergo early onset of hearing loss with strong auditory neuropathy [37], which is contradictory to the strong regeneration of the phoenix spheres. It needs more studies until we can understand the huge difference in the behaviour of the phoenix spheres in different microenvironments. A possible explanation is that a sturdy immune system may have suppressed uncontrollable cell proliferation while the in vitro culture lacks of proper regulators, such as CD4^+^ T cells [38,39]. DNA repair is another critical way that stem cells maintain rapid proliferation and virtually unlimited self-renewal potential. The accumulation of DNA damage can result in loss of genomic integrity and thus remove stem cells from the pluripotent pool [40]. Stem cells rely on pathways that lead to repair of damaged DNA to preserve the integrity of their genomes and get avoid of apoptosis [41]. The enriched GO term category of DNA repair suggests that a robust mechanism of DNA repair may contribute to active regeneration of phoenix spheres. In terms of GSEA, the results further prove that the cell cycle-related gene sets are highly enriched in phoenix spheres. The insignificant enrichment in the auditory, mechanical, and electrical stimulus response gene sets indicate that the phoenix spheres have comparable potential to the mature auditory neurons. The phoenix spheres are capable to be utilised as a substitute of conventional neurons in vitro for various applications. In the PPI network, a total of 10 genes are identified as hub genes, namely Fn1, Gapdh, Mmp9, Vegfa, Itgb3, Cdk1, Col1a1, Met, Fgf2, and Aurka. These genes have intimate connections with cell cycle (Cdk1 and Aurka) and cellular organisation (Fn1, Gapdh, Mmp9, Vegfa, Itgb3, Col1a1, Met, and Fgf2), which have been proved by their enrichment in the top three functional clusters of the PPI network.

Stem cells have unlimited self-renewal capabilities while highly differentiated cells commit to senescence and apoptosis. A group of undifferentiated cells at the mesomeric status between stem cells and terminally differentiated cells are known as transit-amplifying cells (TACs) [42]. Once stem cells exit their quiescent state, they immediately convert into TACs and start rapid mitosis. The cell fate commitment of TACs is an important event in the stem cell homeostasis, which is maintained by precise cell cycle regulation. Mitotic kinases exert pivotal functions throughout mitotic progression [43]. Cdk1 is the most prominent mitotic kinase, which functions in phosphorylation and promotion of mitotic progress [44]. Aurka, a member of the serine/threonine kinase family, is necessary for the division of centrosomes and the production of the mitotic spindle [43]. The Aurka and Cdk1 comprise an important signalling axis in mammal mitosis. In the signalling cascade, Aurka is the regulator at the very upstream and activates Plk1 (non-hub gene, in cluster 1) through phosphorylation, which is regulated by a series of events, and eventually activates the effector Cdk1 via dephosphorylation [45]. Previous studies show that the Aurka-Cdk1 axis is a part of a feedback activation loop at mitosis entry and its activity peaks during mitosis and mitotic exit [46,47,48]. Based on above evidence, the phoenix spheres may belong to the TACs population, hence exhibit active mitosis with involvement of the Aurka-Cdk1 axis.

The remaining seven hub genes are associated with cellular organisation. Among them, four genes Fgf2, Mmp9, Fn1, and Gapdh are incorporated in the cluster 2, which is enriched in BP GO terms of the cytoskeleton. Although the relationship between cytoskeleton structure and stemness remains largely elusive, relaxation of cytoskeleton tension increased expression of pluripotent gene expression [49]. The Fgf2 is reported to lead to cytoskeleton rearrangement since a large number of cytoskeletal proteins and interactors were regulated in phosphorylation, including MAPK/ERK and PI3K/AKT pathways [50]. The Fgf2 is up-regulated in the phoenix spheres and the activation of these pathways is facilitated to the maintenance of stemness [50]. Mmp9 is a metalloproteinase involved in cell attachment. The increased expression of Mmp9 can result in loss of contact with the basement membrane and cytoskeleton rearrangement [51]. Fn1 is a glycoprotein involved in cell attachment, migration, and actin cytoskeleton signalling [52]. In this study, Mmp9 and Fn1 are both significantly down-regulated in phoenix spheres, which indicates an increased cell adhesion and decreased migration. Therefore, it is possible that robust proliferation abilities of the phoenix spheres are a result of low cytoskeleton tension and enhanced cell adhesion. Further studies are needed to fully reveal the underlying mechanisms. Genes Vegfa, Col1a1, and Itgb3 are linked to extracellular matrix (ECM) organisation (cluster 3). As a key component of the niche, ECM serves as a scaffold for cellular support as well as a source of signals that control stem cell self-renewal, differentiation, adhesion, and migration [53,54]. Vegfa is a well-known signalling protein in angiogenesis. Its functions are likely to be context- and cell-type-dependent [55]. It is reported that Vegfa gets involved in endothelial cell migration [56] and stem cell survival [57]. The potential mechanism of Vegfa in the phoenix spheres remains to be characterised. Col1a1 is a subunit of type I collagen. It is one of the main components of ECM. Chen et al. [58] reported that the expression of Col1a1 is critical in maintenance of ECM adhesion and mouse spermatogonia self-renewal. Itgb3 is a subunit of integrin, which is a key candidate in cell–ECM adhesion. By up-regulating the expression of Itgb3 in the phoenix spheres, the cell adhesion to the ECM may be enhanced, and thus regulates self-renewal [59]. In summary, the nearly unlimited self-renewal ability of phoenix spheres may be a result of comprehensive events including activated mitotic signalling pathways, low cytoskeleton tension, and enhanced cell–ECM adhesion. All of these conditions collectively contribute to the homoeostasis and molecular interactions in the phoenix spheres.

Currently, the primary phoenix sphere harvest still relies on animals and needs skilful micromanipulation. Direct reprogramming provides an innovative technology that can convert one cell type into another without turning back to the pluripotent status [60]. Different from the protocol of induced pluripotent stem cells (iPSCs), direct reprogramming bypasses complicated processes of dedifferentiation or redifferentiation, and can manipulate cell fate through some biomolecules, such as TFs. Its effectiveness and efficiency ensure target cells can be produced at a large scale in labs, especially those cells with challenging harvest methods, e.g., neurons. Generally, the cDNA of TFs is integrated into lentiviral constructs and transfected into starting cells by viral infection. In this way, the target cDNAs can be quickly (normally 24 h) integrated into the genome of starting cells and expressed stably. A vast amount of literature has reported successful conversion of various types of starting cells into neurons by transfection of TFs [61,62,63], which proves that the direct reprogramming is a feasible and reliable method to generate neurons in vitro. In this study, we identified 66 potential TFs that can be used in direct reprogramming of phoenix spheres. The top 10 TFs with the highest significance are GKLF, WT1, CPBP, BCL6B, NF-1B, Pax-4, ZIC3, ZF5, LF-A1, and CTCF. They are promising candidates that may be able to convert other somatic cells into phoenix spheres.

With the understanding of the genetic mechanism of self-renewing capability and the predicted TFs for direct reprogramming, in vitro experiments based on auditory phoenix sphere will be carried out to verify our findings in the near future. Once the direct reprogramming method is developed, the process of obtaining phoenix spheres will be largely optimised. The nearly unlimited source of neurons can be beneficial to establishing in vitro neuron study platforms for multiple applications (e.g., cell–device interface, cell co-culture, and gene studies) and more importantly, reduce the use of animals following the 3Rs principles.

Although the development of auditory phoenix sphere could provide almost infinite cell sources for potential AN replacement and restoration, the transplantation is still a long road ahead due to the major technical issue of clinical delivery. The precisely delivery of the AN-like cells into targeted places in the dedicated cochlea seems very difficult. The most commonly used injection technique is perilymphatic transplantation via scala tympani which is believed to reduce the chance of surgical trauma [64,65]. However, there are very few cells that can survive in the scala media with high concentration of potassium. The effective cell migration from perilymphatic space to the targeted damaging position remains undetermined. The cochlear nerve trunk route of transplantation may be the most practical way of cell-based therapy. Some researchers were also interested in delivering cells to Rosenthal’s canal but still found difficulties in the cell migration [66]. Moreover, the functional recovery of the auditory neurons also varies from case to case, due to the variable in damaging area [67]. More effort needs to be put into the improvement of cell survival, migration and selective delivery to the targeted place. We are looking forward to the near future where the auditory phoenix can be effectively used for cell-based ANHL therapy with large-scale generation and low cost.

## 4. Materials and Methods

### 4.1. Dataset Information and Preprocessing

The dataset E-MTAB-9441 (https://www.ebi.ac.uk/arrayexpress/experiments/E-MTAB-9441/ (accessed on 26 June 2022)) was downloaded from the repository ArrayExpress (https://www.ebi.ac.uk/arrayexpress/ (accessed on 26 June 2022)) [68], which contains six paired RNA sequencing (RNA-Seq)-based transcriptomes of differentiated (day 7) phoenix auditory mature neurons and undifferentiated (day 0) phoenix sphere neuroprogenitors at passage 12, 21, and 36, respectively. Samples were paired at passage 12, 21, and 36 forming a triplicate. According to the original paper, although collected at different numbers of passages (from P12 to P36), triplicate samples from undifferentiated and from differentiated cells were well clustered, suggesting a conserved pattern of gene expression with passages. Details regarding cell extraction and preparation for RNA-Seq were elaborated in the source literature by Rousset et al. [30]. The downloaded RAW FASTQ files were then preprocessed on the Galaxy (https://usegalaxy.org/ (accessed on 26 June 2022)) [69,70], a platform integrated with hundreds of packages and tools for analysing RNA-Seq data, to perform quality check and remove poorly or non-expressed genes before proceeding to downstream analyses. The FASTQ files were firstly quality-checked by the FastQC (v. 0.11.9) [71], followed by removing low-expressed genes via the Trimmomatic (v. 0.38.0) [72] using the following thresholds: PE; ILLUMINACLIP: TruSeq2:2:30:10:8; SLIDINGWINDOW: 5:20; LEADING: 3; TRAILING: 3; MINLEN: 50. The filtered data were then loaded onto the HISAT2 [73] to map sequences with the UCSC Mus musculus mm10 reference with an average mapping rate of 88.36%, and generated BAM format files. The sequence reads were counted by featureCounts (v. 2.0.1) [74] which generated a matrix table of counts for the following bioinformatic analyses. The initial number of genes in the set was 24,420, of which 18,880 genes were left after the aforementioned preprocessing.

### 4.2. Identification of Differentially Expressed Genes (DEGs)

The as-obtained TXT format count matrix was uploaded onto the NetworkAnalyst 3.0 (https://www.networkanalyst.ca/ (accessed on 28 June 2022)) [75], a comprehensive gene-centric platform supporting gene expression profiling, biological network analysis and visual exploration. A total of 18,416 (97%) out of 18,880 gene IDs were successfully matched with the webtool internal Entrez ID database. Unannotated genes were filtered out and the log2-counts per million method was applied to the remaining genes for normalisation. After that, the limma package [76] was utilised for identification of DEGs with the thresholds of adjusted *p*-value < 0.05 and |log2FC| > 2.0. A table containing identified DEGs was then generated and downloaded for further analysis.

### 4.3. Gene Ontology (GO) and Kyoto Encyclopedia of Genes and Genomes (KEGG) Pathway Enrichment

GO and KEGG pathway enrichment reveals the most significant terms involved in the interested cells, which helps to understand their characteristics and key information. The Gene Set Enrichment Analysis (GSEA) method was used for GO terms and KEGG enrichment on the NetworkAnalyst 3.0. The term networks were created by the webtool internal visualisation plug-in and the Welch’s *t*-test was applied for the term ranking. GO terms are comprised of biological processes (BP), cellular component (CC), and molecular function (MF).

### 4.4. Gene Set Enrichment Analysis

GSEA (https://www.gsea-msigdb.org/gsea/index.jsp (accessed on 3 July 2022)) was carried out to identify genes associated with four self-defined M5: BP GO collections (cell cycle, auditory stimulus response, mechanical stimulus response, and electrical stimulus response). Four self-identified mouse gene collections were firstly obtained as reference gene sets from the Mouse Gene Set Resources (https://www.gsea-msigdb.org/gsea/msigdb/mouse_geneset_resources.jsp (accessed on 3 July 2022)) on the Molecular Signatures Database (MsigDB, v. 7.5.1, https://www.gsea-msigdb.org/gsea/msigdb/index.jsp (accessed on 3 July 2022)). The words ‘cycle’, ‘auditory’, ‘mechanical’, and ‘electrical’ were used as keywords for corresponding collections in searching. All related gene sets were included while those irrelevant were excluded. The four collections were then imported into the GSEA software (v. 4.2.3, Broad Institute, Cambridge, MA, USA) for further analysis. Gene sets with size < 15 genes or >500 genes were filtered out before analysis. A full list of final gene sets in the collections was shown in Table A1. Genes were ranked by the Signal2Noise method. Gene set permutations were performed 1000 times for each analysis to identify gene sets with significant difference. The normalised enrichment score (NES), nominal *p*-value, and false discovery rate (FDR) q-value worked as indicators of association between sample genomes and the reference gene sets. Gene sets fulfilled |NES| > 1, nominal *p* < 0.01, and FDR q-value < 0.25 at the same time were considered as statistically significant.

### 4.5. Protein–Protein Interaction (PPI) Network Analysis

Proteins coded by their correspondent genes function as groups and have complicated interactions in the biological processes. The PPI network considers genes as functional groups instead of individual elements, which analyses genes in a way that is much closer to the real biological processes. The list of identified DEGs was uploaded onto the STRING database (v. 11.5; https://string-db.org/ (accessed on 8 July 2022)) [77] to recover the association between proteins coded by DEGs and other proteins. A confidence score of >0.4 was defined as significant. Then, the generated interaction data were downloaded and visualised via the Cytoscape software (v. 3.9.0, Cytoscape Consortium, San Diego, CA, USA) [78]. The plug-in cytoHubba was utilised to screen the top 10 hub genes ranked by degree, which was defined as the total number of connections between adjacent proteins.

### 4.6. Subcluster Analysis and Transcription Factors (TFs) Prediction

In order to analyse the different functional clusters constituted by proteins in the PPI network, another plug-in Molecular Complex Detection (MCODE) was utilised to extract sub-clusters within the network. Degree cutoff = 2, node score cutoff = 0.2, k-core = 2, and max depth = 100 were set as thresholds. The identified top 5 sub-clusters were input into the g:GOSt on g:Profiler (http://biit.cs.ut.ee/gprofiler/ (accessed on 8 July 2022)) [79] for GO term and KEGG pathway enrichment analysis. The tailor-made g:SCS algorithm [79] and *p* < 0.05 were set as cut-off criteria. Similarly, all nodes in the main PPI network were input into the g:GOSt for TFs prediction using the TRANSFAC database, which is an encyclopedia of eukaryotic transcriptional regulation and a tool to identify potential TF binding sites (TFBSs) [80]. A full list of software and websites used in this study is shown in Table A2.

## 5. Conclusions

In this study, ten hub genes (Fn1, Gapdh, Mmp9, Vegfa, Itgb3, Cdk1, Col1a1, Met, Fgf2, and Aurka) were identified. Together with activated mitotic signalling pathways, low cytoskeleton tension, and enhanced cell–ECM adhesion, the phoenix spheres exhibit nearly unlimited self-renewal characteristics. Ten TFs (GKLF, WT1, CPBP, BCL6B, NF-1B, Pax-4, ZIC3, ZF5, LF-A1, and CTCF) were predicted as promising direct reprogramming factors, which provided valuable information for further screening and optimising of TF combinations. The gene set enrichment analysis proves that the phoenix spheres have full potential of auditory neuron differentiation even at their progenitor status. The robustness and flexibility of the phoenix sphere make it an ideal cell source to a wide range of hearing research and pave the way to a more viable cell-based therapy for hearing loss.

## Figures and Tables

**Figure 1 ijms-23-10287-f001:**
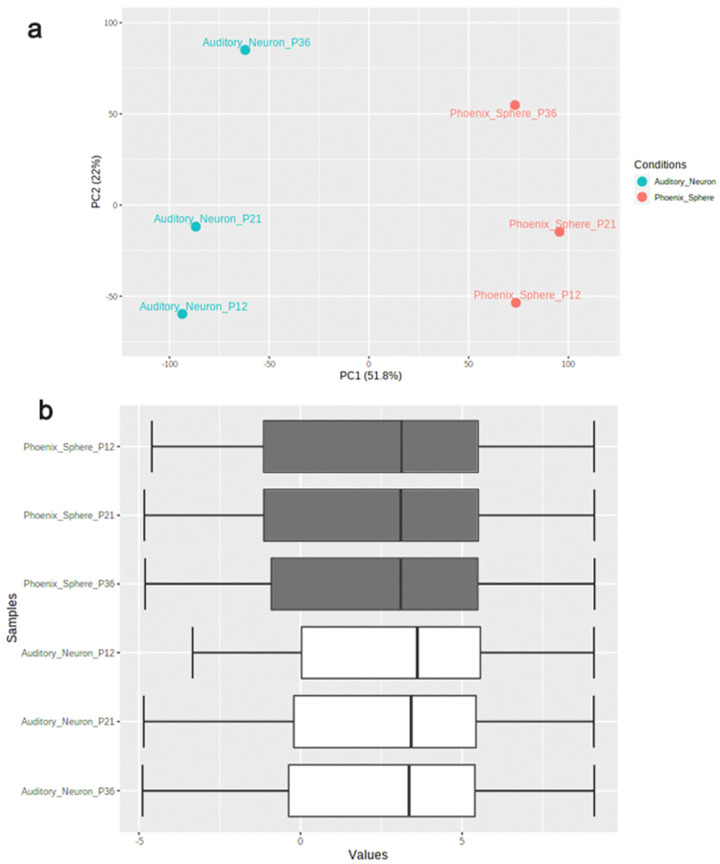
Data quality control of phoenix sphere and auditory neuron samples. (**a**) PCA plot displays that the phoenix sphere and auditory neuron samples were clearly divided into two clusters; (**b**) the quantile normalisation boxplot of samples. Vertical black lines in the boxes represent medians.

**Figure 2 ijms-23-10287-f002:**
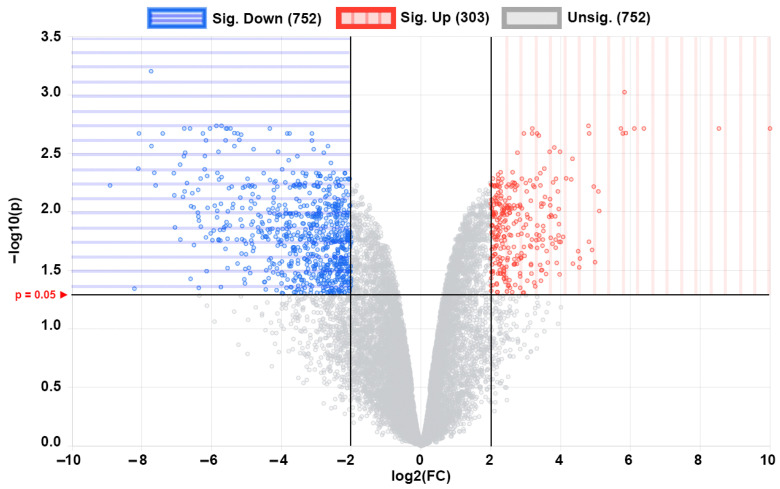
Volcano plot of all 1055 DEGs of phoenix spheres compared to auditory neurons. Each dot represents a gene. Blue area highlights 752 down-regulated DEGs while red area contains 303 up-regulated DEGs.

**Figure 3 ijms-23-10287-f003:**
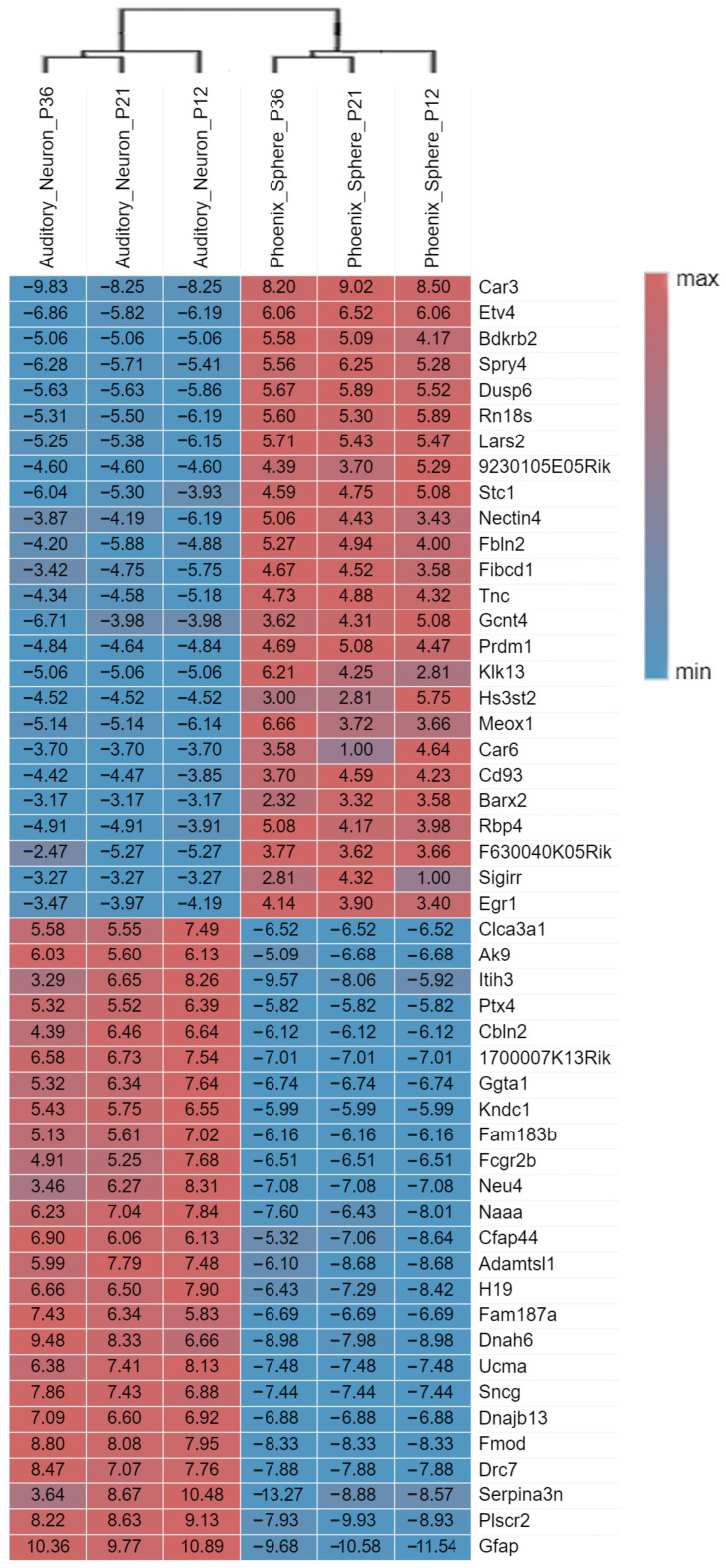
Heatmap of the top 25 up- and down-regulated phoenix sphere DEGs with the highest fold change. The fold change value of each gene is shown in boxes. Red and positive values indicate the fold change of up-regulation while blue and negative values stand for down-regulation. The hierarchical clustering analysis further proves the samples are keeping consistent quality within the group. Each gene’s ‘fold change’ is calculated by comparing a target gene count to the average gene count of the references.

**Figure 4 ijms-23-10287-f004:**
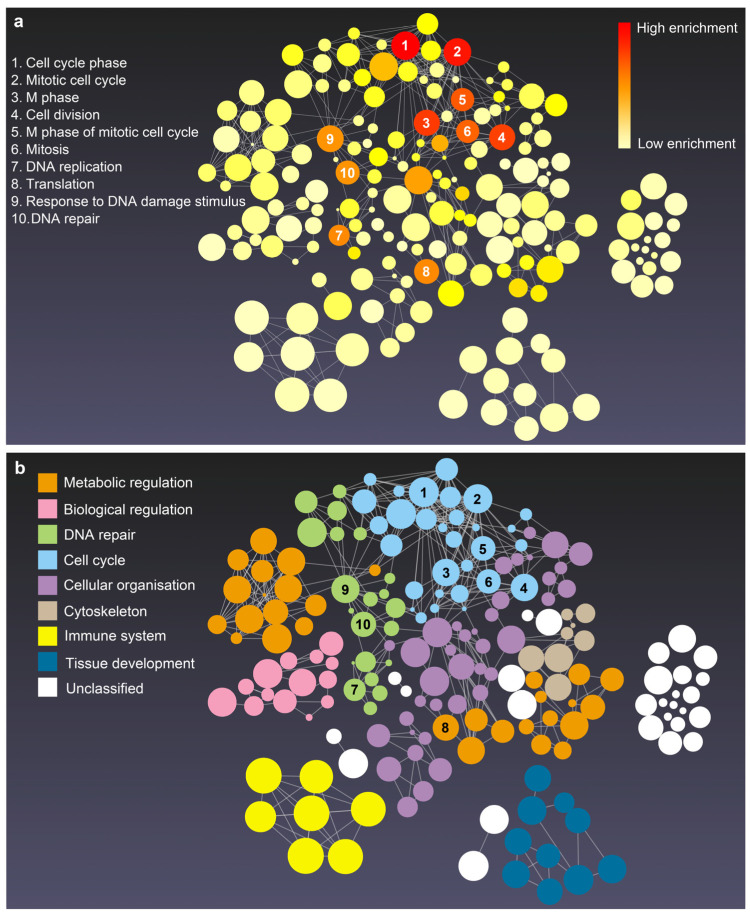
The enrichment network of biological process of phoenix spheres. Created in the enrichment visualisation tool on the NetworkAnalyst platform. Bubble size is associated to the size of gene sets. (**a**) Enrichment network with red to light yellow colours based on the adjusted *p*-value. Red represents the lowest *p*-value while light yellow stands for the highest *p*-value; (**b**) the identical network as (**a**) but coloured according to the categories of GO terms. Each colour belongs to one category of terms. The spatial distance between different categories indicates similarity. The closer the more similar in biological process. The numbers 1 to 10 on (**a**,**b**) represent the top 10 enriched terms.

**Figure 5 ijms-23-10287-f005:**
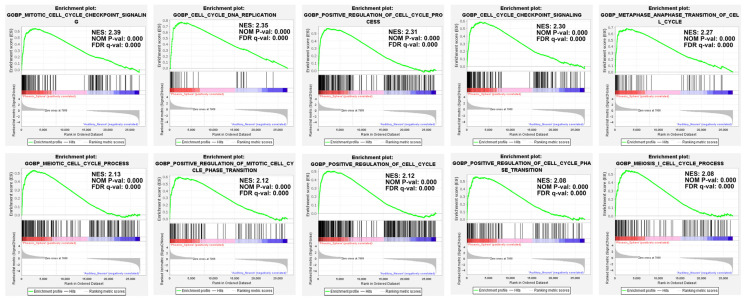
GSEA enrichment plot. In the phoenix sphere group, the top 10 significantly up-regulated gene sets from the cell cycle collection were demonstrated. The mitotic cell cycle checkpoint signalling, cell cycle DNA replication, and positive regulation of the cell cycle process are the leading gene sets.

**Figure 6 ijms-23-10287-f006:**
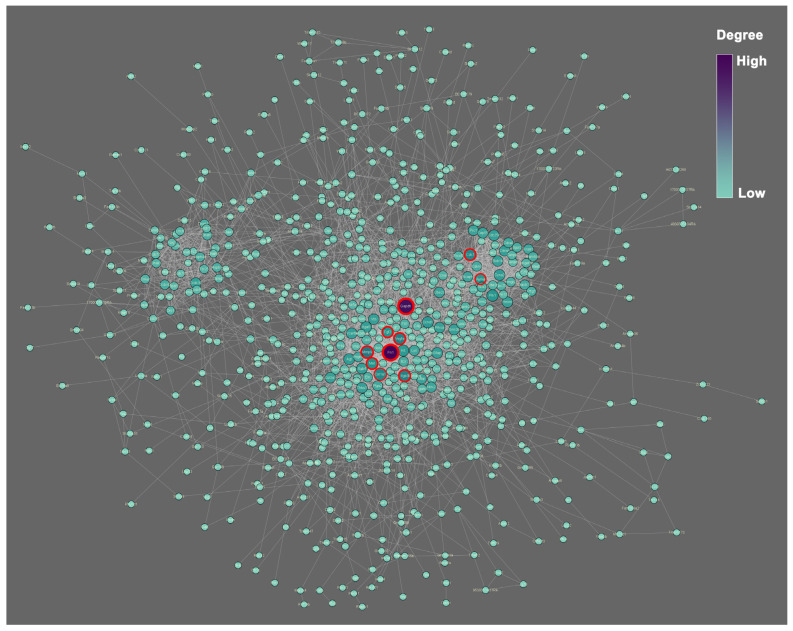
PPI network of all DEGs. The colour scale and node size indicate the connection degree. Purple and larger size represent higher degree, and vice versa. Red circles highlight the top 10 hub genes calculated by the plug-in MCODE on the Cytoscape.

**Figure 7 ijms-23-10287-f007:**
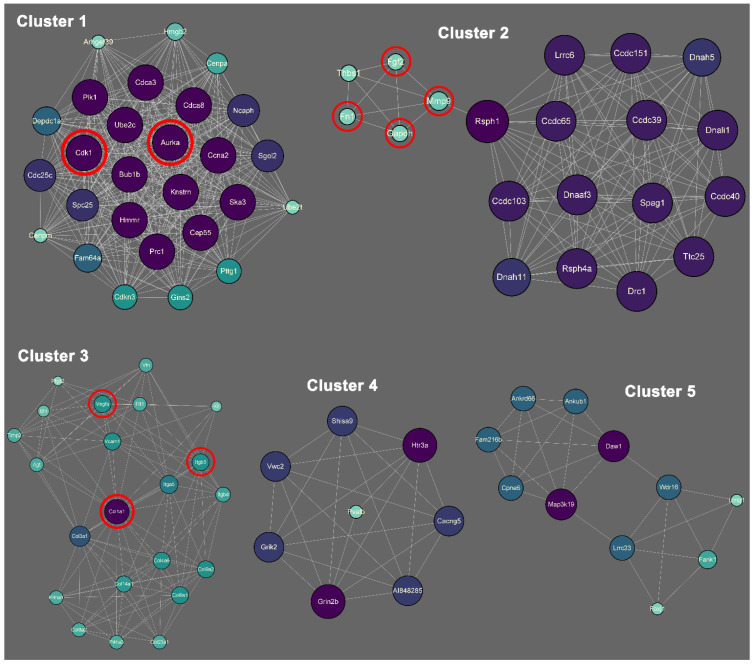
Biofunctional sub-clusters extracted from the main PPI network. Purple and larger size represent higher degree, while green and smaller size indicate lower degree. Red circles highlight the hub genes.

**Figure 8 ijms-23-10287-f008:**
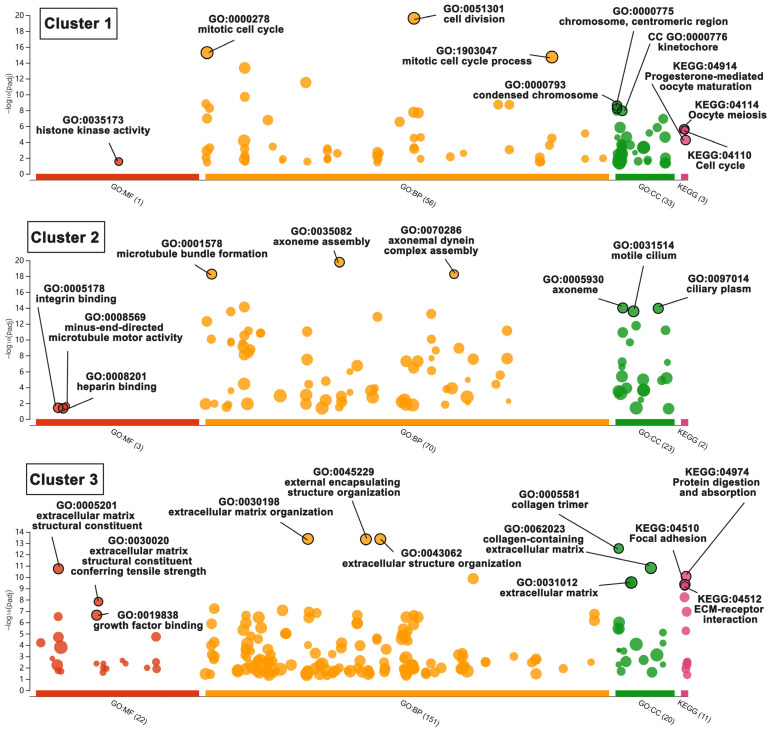
Bubble plot of enriched BP, CC, MF GO terms and KEGG pathways in sub-cluster 1, 2, and 3. Bubble size represents term size. In the vertical direction, the higher the bubble, the more significantly enriched. In the horizontal direction, colour stands for the categories of terms, and distance between bubbles indicates the similarity. Terms from the same subtree stay close to each other.

**Figure 9 ijms-23-10287-f009:**
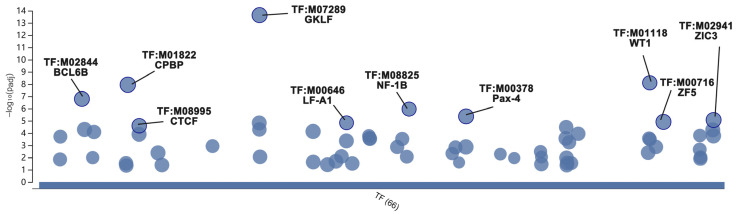
Bubble plot of prediction of key TFs. In the vertical direction, the higher the bubble, the more significantly enriched. In the horizontal direction, TFs within the same subtree stay close to each other.

**Table 1 ijms-23-10287-t001:** Top 5 significantly enriched GO terms and KEGG pathways of DEGs.

Category	GO ID	Description	Gene Count	Adjusted *p*-Value
BP	GO:0022403	Cell cycle phase	474	1.05 × 10^−41^
BP	GO:0000278	Mitotic cell cycle	427	2.08 × 10^−39^
BP	GO:0000279	M phase	348	9.84 × 10^−36^
BP	GO:0051301	Cell division	349	7.86 × 10^−34^
BP	GO:0000087	M phase of mitotic cell cycle	248	6.34 × 10^−32^
CC	GO:0005694	Chromosome	443	5.70 × 10^−48^
CC	GO:0098687	Chromosomal part	380	2.69 × 10^−45^
CC	GO:1990904	Ribonucleoprotein complex	413	4.44 × 10^−36^
CC	GO:0044391	Ribosomal subunit	111	2.25 × 10^−34^
CC	GO:0005840	Ribosome	160	8.52 × 10^−32^
MF	GO:0003735	Structural constituent of ribosome	102	7.22 × 10^−27^
MF	GO:0003677	Structure_specific DNA binding	159	1.84 × 10^−10^
MF	GO:0005198	Structural molecule activity	279	2.09 × 10^−9^
MF	GO:0003682	Chromatin binding	255	7.07 × 10^−7^
MF	GO:0043565	Sequence_specific DNA binding	451	7.07 × 10^−7^
KEGG	KEGG:03010	Ribosome	114	1.32 × 10^−34^
KEGG	KEGG:04110	Cell cycle	100	2.47 × 10^−13^
KEGG	KEGG:03030	DNA replication	33	7.25 × 10^−12^
KEGG	KEGG:14543	RNA transport	117	2.75 × 10^−9^
KEGG	KEGG:03440	Homologous recombination	34	2.63 × 10^−7^

**Table 2 ijms-23-10287-t002:** The top 10 hub genes ranked by the connection degree.

Rank	Gene symbol	Degree
1	Fn1	96
2	Gapdh	91
3	Mmp9	57
4	Vegfa	54
5	Itgb3	53
6	Cdk1	51
7	Col1a1	50
8	Met	47
9	Fgf2	46
10	Aurka	44

**Table 3 ijms-23-10287-t003:** Node, edge, and score of the top 5 sub-clusters.

Cluster No.	Node	Edge	Score
1	27	329	25.31
2	20	115	12.11
3	22	115	10.95
4	8	23	6.57
5	11	26	5.20

**Table 4 ijms-23-10287-t004:** Top three enriched BP, CC, MF, and KEGG terms in the sub-cluster 1, 2, and 3.

Cluster No.	Category	GO ID	Description	Gene Count	Adjusted *p*-Value
1	BP	GO:0051301	cell division	17	2.42 × 10^−20^
	BP	GO:0000278	mitotic cell cycle	16	5.22 × 10^−16^
	BP	GO:1903047	mitotic cell cycle process	15	1.87 × 10^−15^
	CC	GO:0000793	condensed chromosome	9	2.50 × 10^−9^
	CC	GO:0000775	chromosome, centromeric region	8	6.55 × 10^−9^
	CC	GO:0000776	kinetochore	7	8.56 × 10^−9^
	MF	GO:0035173	histone kinase activity	2	2.86 × 10^−3^
	KEGG	KEGG:04114	Oocyte meiosis	5	2.31 × 10^−6^
	KEGG	KEGG:04110	Cell cycle	5	2.63 × 10^−6^
	KEGG	KEGG:04914	Progesterone-mediated oocyte maturation	4	5.53 × 10^−5^
2	BP	GO:0035082	axoneme assembly	11	1.68 × 10^−20^
	BP	GO:0001578	microtubule bundle formation	11	5.38 × 10^−19^
	BP	GO:0070286	axonemal dynein complex assembly	9	5.50 × 10^−19^
	CC	GO:0005930	axoneme	9	9.85 × 10^−15^
	CC	GO:0097014	ciliary plasm	9	1.13 × 10^−14^
	CC	GO:0031514	motile cilium	10	2.66 × 10^−14^
	MF	GO:0008569	minus-end-directed microtubule motor activity	2	2.27 × 10^−2^
	MF	GO:0005178	integrin binding	3	3.61 × 10^−2^
	MF	GO:0008201	heparin binding	3	4.56 × 10^−2^
3	BP	GO:0030198	extracellular matrix organization	11	4.28 × 10^−14^
	BP	GO:0045229	external encapsulating structure organization	11	4.45 × 10^−14^
	BP	GO:0043062	extracellular structure organization	11	4.61 × 10^−14^
	CC	GO:0005581	collagen trimer	8	3.00 × 10^−13^
	CC	GO:0062023	collagen-containing extracellular matrix	10	1.56 × 10^−11^
	CC	GO:0031012	extracellular matrix	10	3.00 × 10^−10^
	MF	GO:0005201	extracellular matrix structural constituent	8	1.82 × 10^−11^
	MF	GO:0030020	extracellular matrix structural constituent conferring tensile strength	5	1.48 × 10^−8^
	MF	GO:0019838	growth factor binding	6	2.30 × 10^−7^
	KEGG	KEGG:04974	Protein digestion and absorption	9	8.57 × 10^−11^
	KEGG	KEGG:04510	Focal adhesion	10	4.37 × 10^−10^
	KEGG	KEGG:04512	ECM-receptor interaction	8	6.44 × 10^−10^

**Table 5 ijms-23-10287-t005:** Predicted top 10 key TFs.

TF ID	TF Name	Adjusted *p*-Value
TF:M07289	GKLF	2.34 × 10^−14^
TF:M01118	WT1	7.90 × 10^−9^
TF:M01822	CPBP	1.14 × 10^−8^
TF:M02844	BCL6B	1.64 × 10^−7^
TF:M08825	NF-1B	1.11 × 10^−6^
TF:M00378	Pax-4	4.39 × 10^−6^
TF:M02941	ZIC3	8.58 × 10^−6^
TF:M00716	ZF5	1.23 × 10^−5^
TF:M00646	LF-A1	1.46 × 10^−5^
TF:M08995	CTCF	2.56 × 10^−5^

## Data Availability

The gene expression profile E-MTAB-9441 can be downloaded from a public functional genomics data repository ArrayExpress database (https://www.ebi.ac.uk/arrayexpress/, accessed on 26 June 2022).

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
