# Peer review of "Genetic Mechanism Study of Auditory Phoenix Spheres and Transcription Factors Prediction for Direct Reprogramming by Bioinformatics"

_ijms, 2022, doi:10.3390/ijms231810287_

Round 1

Reviewer 1 Report

In my opinion, it is a very good work, which opens new perspectives not only in hearing pathology, but also possibly in other neural pathologies. My recommendation is that after this beautiful theoretical study, practical applications should be carried out.

Author Response

We would thank the reviewers for their positive comments and kind suggestions. The additional details have been included and highlighted in yellow in the revised manuscript.

We agree that further experiments will need to be carried out to investigate the potential of the phoenix sphere in practical application. Based on the understanding of the genetic mechanism of self-renewing capability and the predicted TFs for direct reprogramming, in vitro experiments have been planned in our lab. The relative discussion has been added on Page 16 Lines 326-332.

Reviewer 2 Report

Chen et al performed a bioinformatic analysis based on RNA seq samples of differentiated and undifferentiated phoenix cells to find differential expressed genes (DEG) and identify hubs and potential transcription factors (TF) of interest. They used bioinformatic tools widely accepted such as Gene ontology, transfact and GSEA and identify 10 hubs and 10 interesting TFs. Although this work is novel and the genes identified maybe will be of interest for the community, it is not clear how authors used the datasets because they were derived from samples from different time points. In addition, it is not clear if some of the genes of ther list has previously shown as DEG in these cells or if have been studied in this context.

Major points:

Are RNAseq dataset (differentiated and undifferentiated phoenix cells) considered triplicates even when they are different time points? Or they were used as samples without replicates? Limma framework requires replicates? Authors should mention these points and justify their use as replicate samples if that is the case. Considering all the work is based on these DEGs, this is a crucial point.

For DEGs found in this work, some of these genes have been previously shown as DEG in these experimental set up? This would be an idea of the validity of the comparison

In figure 3, numbers of the heat map are "fold change" of each gene in each library, but it is not clear what is the comparison? fold change to some control gene?

some of TF identified in this work has been previously used in other experimental model for direct reprogramming?

Minor points

In Results section, a brief sentence of the samples/libraries is suggested to start the description.

Line 57, typo "nan-autologous"

Author Response

Dear reviewer, 

Thanks for your comments and they are very helpful. The manuscript has been revised accordingly (highlighted in the revised manuscript):

1. The RNA-seq samples are paired samples to form triplicate. Here attached figure (Figure 3C from original paper) and explanation from the original paper (doi: 10.3389/fncel.2020.599152):
'Although collected at different numbers of passages (from P12 to P36), triplicate samples from neuroprogenitors and from differentiated cells were well clustered, suggesting a conserved pattern of gene expression with passages. Similarly, the heatmap representation of differentially expressed genes revealed comparable patterns of gene expression between triplicates.'

2. According to the 'Limma: linear Models for Microarray Data' written by G. K. Smyth, limma uses linear models to analyze designed microarray experiments. This approach allows very general experiments to be analyzed nearly as easily as a simple replicated experiment. The approach requires a minimum of two matrices to be specified. The first is the design matrix which provides a representation of the different RNA targets which have been hybridized to the arrays. The second is the contrast matrix which allows the coefficients defined by the design matrix to be combined into contrasts of interest. Each contrast corresponds to a comparison of interest between the RNA targets. For very simple experiments the contrast matrix may not need to be specified explicitly.

3. Compared to the original report, although we used a narrower threshold to identify DEGs, our findings validated the previous report. For example, the GO term of cell cycle and DEGs of Fa2h, Itgb4, Lgi4, Ntrk2, Miat, and so forth were also enriched in our data. These points have been added to the manuscript.

4. In Figure 3, each gene's 'fold change' is calculated by comparing a target gene count to the average gene count of the references. E.g., gene A, sample 1 compare to gene A, average count of sample 4, 5, and 6, vice versa.

5. Some TFs (e.g. WT1, Pax-4, ZIC3, and CTCF) identified in this study have been used for direct reprogramming to other cell types. The critical point is the combination of TFs. A specific combination may make TFs function in a different way and thus lead to a different cell fate. This screening work has been planned to be carried out in the near future.

6. A brief sentence of the sample information for downstream analysis has been added.

7. Line 57, the typo has been corrected.
